# Women in Advanced Reproductive Age: Are the Follicular Output Rate, the Follicle-Oocyte Index and the Ovarian Sensitivity Index Predictors of Live Birth in an IVF Cycle?

**DOI:** 10.3390/jcm11030859

**Published:** 2022-02-06

**Authors:** Andrea Roberto Carosso, Rik van Eekelen, Alberto Revelli, Stefano Canosa, Noemi Mercaldo, Chiara Benedetto, Gianluca Gennarelli

**Affiliations:** 1Obstetrics and Gynecology 1U, Physiopathology of Reproduction and IVF Unit, Sant’Anna Hospital, Health and Science City, University of Torino, Via Ventimiglia 3, 10126 Turin, Italy; s.canosa88@gmail.com (S.C.); noemimercaldo@gmail.com (N.M.); chiara.benedetto@unito.it (C.B.); gennarelligl@gmail.com (G.G.); 2Centre for Reproductive Medicine, Academic Medical Centre, Amsterdam UMC, 1105 AZ Amsterdam, The Netherlands; r.vaneekelen@amsterdamumc.nl

**Keywords:** IVF, unexplained infertility, follicular output rate, follicle-oocyte index, ovarian sensitivity index, ovarian responsiveness, live birth

## Abstract

(1) **Background**: Several researchers have investigated alternative markers related to ovarian responsiveness in order to better predict IVF outcomes, particularly in advanced reproductive-aged women. The follicular output rate (FORT), the follicle-oocyte index (FOI) and the ovarian sensitivity index (OSI) are among the most promising. However, these three metrics have not been investigated as independent predictors of live birth in women of advanced reproductive age; neither have they been compared to the two ‘component’ characteristics that are used to calculate them. (2) **Methods**: A logistic regression model containing all relevant predictors of ovarian reserve or response was used to evaluate the potential of FORT, FOI and OSI as predictors of live birth. After, the non-linear associations between FORT, FOI and OSI and the probability of live birth were evaluated. Finally, we fitted multiple logistic regression models to compare whether FORT, FOI and OSI were more informative predictors than their components. (3) **Results**: 590 couples received a total of 740 IVF cycles, after which, 127 (17.5%) obtained a live birth. None of FORT, FOI and OSI showed a strength of association or a p-value even close to female age (odds ratio for live birth (95% confidence interval) 1.00 (0.99–1.01), 1.00 (0.99–1.01), 0.98 (0.88–1.11) and 0.58 (0.48–0.72), respectively). The three models comparing FORT, FOI and OSI with the number of oocytes retrieved, the AFC, the number of preovulatory follicles and the FSH total dose were not more informative. (4) **Conclusions**: In a population of women of advanced age with unexplained infertility, none of FORT, FOI and OSI were predictive of live birth or more predictive than the two ‘component’ characteristics that were used to calculate them. We suggest clinicians and researchers still use female age as the most reliable predictor of an IVF treatment.

## 1. Introduction

The number of retrieved oocytes is considered one of the most important predictors of live birth in an in vitro fertilization (IVF) cycle [1,2,3]. Oocyte yield depends on several factors that modulate the ovarian responsiveness to controlled ovarian stimulation (COS). These include the type and dose of exogenous gonadotropins; the intrinsic sensitivity of the ovary to hormonal stimulation, partially correlated to the polymorphic variants of the follicle-stimulating hormone (FSH)-receptor [4]; and the rhythm of follicular maturation waves [5]. In particular, variants in genes of FSH β-chain (FSH-B) and its receptor (FSH-R) seem to be the most promising candidates for a pharmacogenomic approach to controlled ovarian stimulation in assisted reproductive technologies [6].

The individual variability of these intrinsic factors may be associated with an unexpectedly low ovarian response during COS. Thus, the total number of retrieved oocytes does not always accurately reflect the ovarian potential. The wish to find the true ovarian potential in terms of the maximum number of retrieved oocytes and to link this to aforementioned factors has stimulated the research of more sophisticated markers of ovarian function and response, such as the ovarian sensitivity index (OSI) [7,8], the follicular output rate (FORT) [9,10] and the follicle-oocyte index (FOI) [11]. The OSI is defined as the ratio between the number of retrieved oocytes and the total dose of FSH administered; the FORT represents the ratio between the number of pre-ovulatory follicles obtained after COS and the pre-stimulation pool of small antral follicles; and the FOI is the ratio between the number of retrieved oocytes and the number of antral follicles at the beginning of COS.

OSI represents ovarian sensitivity to exogenous gonadotropins, and since its introduction, it has been used to adjust the COS regimen in subsequent IVF cycles, whereas FORT and FOI have been claimed to be a good representations of the dynamic nature of follicular growth and follicular competence. As a matter of fact, several studies have shown that these markers correlate positively with IVF outcome [9,12,13].

However, the aforementioned markers are also affected by some limitations: it is unlikely that all three metrics have simple linear associations with the probability of live birth. OSI does not take into account the gonadotropin regimen, nor does it consider the type of gonadotropin (recombinant, urinary, with luteinizing hormone (LH)-like activity, or not). Furthermore, it might be misleading if inappropriate low starting doses of exogenous gonadotrophins are given. Neither FORT nor FOI assess the total number of mature oocytes retrieved, whereas both indexes depend on the baseline antral follicle count (AFC). AFC by itself shows limited predictive value in older women, whose follicles are affected by increased granulosa cell apoptosis, impaired mitochondrial function and increased oxidative stress [14,15]. Indeed, older women represent the class of patients in whom it is more difficult to predict the chances of IVF success. In this context, the use of multiple surrogate markers that indicate ovarian reserve from different perspectives appears promising. 

However, no studies have so far considered these three metrics (FORT, FOI and OSI) as independent predictors and compared metrics to the two ‘component’ characteristics that were used to calculate them. In this retrospective study among women aged 39 or above, we first evaluated non-linear associations between metrics and live birth in women of advanced reproductive age, then compared all three metrics to one another, and finally, compared these three metrics to the two variables that were used to calculate them.

## 2. Materials and Methods

### 2.1. Patient Population

The study was performed in accordance with the Helsinki Declaration and with the approval of the Institutional Review Board. In this retrospective analysis, we included a cohort of women aged 39 years or above and affected by unexplained infertility, who were admitted to the IVF unit of the S. Anna academic hospital between 2010 and 2019. Only autologous cycles were considered. Exclusion criteria were: female body mass index (BMI: Kg/m^2^) > 32 kg/m^2^; anti-Mullerian hormone (AMH) < 0.1 and/or early follicular phase follicle-stimulating hormone (FSH) > 20 UI/l; any known cause of female infertility (i.e., previous history of pelvic inflammatory disease, positive anti-Chlamydia IgG, endometriosis, anovulation, etc.); and cycles with pre-implantation genetic diagnosis. The BMI criterion is imposed by regional legislation in order to reduce the risk of comorbidities related to obesity in pregnancy. All women included in the study had ovulatory cycles and patency of at least one fallopian tube at sonosalpingography (SSG), and all male partners had normal basic semen parameters according to the indications of the World Health Organization (WHO), 2010.

### 2.2. ART Procedure

Controlled ovarian stimulation (COS) was performed either with recombinant FSH (rFSH), human menopausal gonadotropin (HP-hMG) or rFSH plus recombinant luteinizing hormone (rLH), under pituitary suppression applying both long and short protocols. The choice of the starting gonadotropin dose was based on age, AMH concentrations and AFC, as well as on response to previous COS. The long protocol was performed by administering buserelin (Suprefact, Hoechst, Frankfurt, Germany; 900 mcg/d intranasally) starting from the late luteal phase of the previous cycle. Pituitary suppression was verified after approximately two weeks (appearance of a menstrual bleeding, serum estradiol <50 pg/mL, endometrial thickness <3 mm) before starting COS. In the short protocol, either the GnRH-antagonist cetrorelix (Cetrotide, Merck, Darmstadt, Germany) or ganirelix (Orgalutran fi, Merck Sharp & Dome, Kenilworth, NJ, USA) was started at a subcutaneous dose of 0.25 mg/d according to a flexible schedule, when at least one follicle ≥12 mm in mean diameter was observed on ultrasound (US).

COS was monitored by serial transvaginal US and serum estradiol (E2) measurements performed every second day from stimulation day 6–7. COS continued until at least two follicles reached 18 mm in mean diameter, when ovulation was triggered by injecting either 10,000 international units (IU) of human chorionic gonadotropin (hCG) (Gonasi HPfi, IBSA, Lugano, Switzerland) or 250 mcg of rhCG (Ovitrellefi, Merck, Darmstadt, Germany) subcutaneously. US-guided transvaginal oocyte aspiration (OPU) was performed approximately 36–37 h after hCG administration, under local anesthesia (paracervical block). Oocytes were immediately recovered from the follicular fluid and then washed in buffered medium and stored until the fertilization procedure.

Semen samples were examined to assess sperm concentration, motility and morphology according to the WHO guidelines [16]. The samples were then prepared by density gradient centrifugation in order to select motile, morphologically normal spermatozoa. Conventional IVF or ICSI were performed on all available oocytes within 4 h of oocyte collection or 2 h after cumulus cell removal, respectively. After 16–18 h of incubation in a controlled atmosphere, the occurrence of normal fertilization was assessed.

### 2.3. Embryo Selection and Transfer

Zygotes were placed in pools in 4-well dishes (Thermo Scientific, Roskilde, Denmark), and embryos were cultured in pre-equilibrated Cleavage Medium (Cook) overlain with mineral oil using incubators (Minc, COOK, Bloomington, IN, USA) in a gas phase consisting of 5% O_2_, 6–7% CO_2_, balanced with N_2_.

All cleaved embryos were morphologically evaluated under a conventional stereomicroscope using the integrated morphology cleavage score (IMCS) by Holte [17].

Embryo(s) transfer was performed using a soft catheter under ultrasound guidance. According to the policy of our IVF unit during the time period under study, one or two embryos were transferred on day 2/3 post-fertilization. Only embryos reaching the blastocyst stage were vitrified and thawed in subsequent cycles. The luteal phase was supported by administering 180 mg/day natural progesterone (Crinone 8fi, Merck, Darmstadt, Germany) for 15 days.

Pregnancy was assessed by serum hCG assay 15 days after embryo transfer (ET) and then confirmed if at least one gestational sac was visualized on transvaginal US after two further weeks.

Live birth was defined as the delivery of a live-born infant (>24 weeks of gestation). The cumulative live birth rate (CLBR) was defined as live deliveries (at least one live birth) per women including both fresh and frozen/thawed embryo transfers obtained from the same cycle.

### 2.4. Metrics

FORT was defined as the number of pre-ovulatory follicles of at least 15 mm in response to ovarian stimulation divided by the pre-treatment AFC (which included all follicles at least 11 mm in mean diameter) multiplied by 100. 

FOI was defined as the number of oocytes collected after ovarian stimulation divided by the pre-treatment AFC multiplied by 100. For agonist cycles, the AFC was evaluated as previously described [9].

OSI was defined as the number of oocytes collected after ovarian stimulation divided by the total dose of FSH administered, per 1000.

### 2.5. Statistical Analyses

First, we fitted a logistic regression model containing all relevant predictors of ovarian reserve or response to evaluate whether these common indicators were ‘independent’ predictors of live birth: female age, duration of infertility, previous assisted reproduction treatment (ART), primary or secondary subfertility, preovulatory follicles, oocytes collected, AFC and total FSH dose. Since only age was found to be a strong predictor of live birth, we evaluated the non-linear associations between the three metrics separately (FORT, FOI and OSI) and the estimated probability of live birth after IVF to obtain an indication of the strength and functional form of the associations. To this end, we fitted logistic regression models on live birth after IVF using restricted cubic splines with 4 knots for the three mentioned metrics and then used these models to predict the probability of a live birth after IVF for a range of values for each metric. We plotted the results using simultaneous 95% confidence intervals (Cis) that take into account the many comparisons drawn in these plots.

Next, we fitted multiple logistic regression models to compare whether FORT, FOI and OSI were more informative predictors than the two ‘components’ used to calculate them. For FORT, we compared a model containing FORT to a model containing preovulatory follicles and AFC. For FOI, we compared a model containing FOI to a model containing oocytes collected and AFC. For OSI, we compared a model containing OSI to a model containing oocytes collected and total FSH dose. In addition, the following four factors were added to all models, as they were considered the most important and could explain part of the associations of the three metrics and/or their components: female age, duration of infertility, previous ART and primary or secondary subfertility.

We also assessed whether the associations between FORT, FOI and OSI were different for poor responders to stimulation (according to Bologna criteria [18] defined as AMH < 1.1 or AFC < 7) by including an interaction between the predictor and a factor denoting whether a woman was a poor responder.

We determined the best-fitting and most-informative model in terms of the lowest Akaike Information Criterion (AIC) [19]. For all models, we used a robust standard error that allowed for the clustering of patient identities to adjust precision, since couples could receive multiple cycles [20].

### 2.6. Sample Size Calculation and Software

Using the ‘10 events per variable’ rule of thumb, we would be able to include approximately 11–13 predictors in our model(s) [21]. Using a more elaborate, contemporary method, we calculated that, with 10 candidate predictors and approximately 18% having the event, 700 participants were required to obtain an accuracy of 0.05 in the mean average percentage error [22].

Data were compiled in Excel and analyzed in R 3.6.0 using the rms, mice, xtable and dplyr packages (R Core Team, Vienna, Austria, 2017).

## 3. Results

Data on 590 couples were available. These 590 couples received a total of 740 IVF cycles, after which 127 (17.5%) obtained a live birth. The baseline characteristics are shown in Table 1.

In Table 2, we report the fully adjusted model, showing associations between patient characteristics and the odds of a live birth after IVF. After adding all predictors, only female age (*p* < 0.001) was a significant predictor of live birth after IVF (Table 2). None of the metrics (FORT, FOI and OSI) showed a strength of association or a *p*-value even close to female age, even considering the different scales they are on.

### 3.1. Non-Linear Associations between Metrics and Live Birth after IVF

Figure 1, Figure 2 and Figure 3 show the non-linear associations between live birth and FORT, FOI and OSI, respectively. 

For all metrics, we see that higher values were associated with a higher estimated probability of live birth after IVF. For FORT and FOI, this seems to be somewhat linear over the range of their possible values from 0 to 100, but for OSI, the association seems to be non-linear. We therefore decided to use the non-linear fit for the comparison with its two components in the next analysis.

### 3.2. Comparing Metrics to Their Two Components

The model with FORT was not more informative than the model with preovulatory follicles and AFC, as shown by a 5-point-higher AIC for the former. The model with FOI was not more informative than the model with oocytes collected and preovulatory follicles, as shown by a 5-point-higher AIC for the former. The model with OSI modelled as non-linear was not more informative than the model with oocytes collected and FSH dose, as shown by a 4-point-higher AIC for the former.

The associations between live birth and FORT, FOI and OSI were not significantly different in poor responders than in normal responders (*p*-values for interactions 0.744, 0.151 and 0.995).

## 4. Discussion

In this retrospective analysis, the predictive value for obtaining a live birth of three IVF indicators was evaluated in elderly women of couples affected by unexplained infertility. FOI, FORT and OSI do not show a stronger or more informative association with live birth than the components used for their calculation, i.e., the number of oocytes retrieved, the AFC, the number of preovulatory follicles and the FSH total dose. Female age remained the most reliable predictor for live birth in an IVF cycle.

Women of advanced reproductive age remain an open dilemma and a challenge for all clinicians working in the field of assisted reproductive technologies (ART). 

Over the years, several attempts have been made in order to identify surrogate markers of ovarian reserve, which, in turn, could be markers of IVF outcomes. This would allow us to decide for whom treatment is expected to be (cost-)beneficial. However, so far, studies on the association between markers such as AMH/AFC and implantation, pregnancy, and/or live birth after assisted conception have reported conflicting results [23,24]. 

Therefore, several authors have investigated alternative markers that are discussed in this paper, particularly those related to ovarian responsiveness to COS, in order to better predict IVF outcomes. Efficient markers would be of particular interest to subgroups of low prognosis patients defined both by the Poseidon and the Bologna criteria [18,25]. 

Alviggi et al. analyzed the predictive role of FOI in assessing ovarian sensitivity in hypo-responder patients [11]. The authors concluded that FOI might reflect the dynamics of follicular growth in response to exogenous gonadotropin better than traditional markers of ovarian reserve. In particular, low FOI values imply that only a fraction of available antral follicles were exploited during COS, suggesting that there might be therapeutic opportunities (increasing the FSH dose and/or adding LH) to improve ovarian responsiveness, and therefore, the overall prognosis.

Grynberg et al. discussed the potential use of FORT as a quantitative and qualitative marker of ovarian responsiveness to gonadotropins, and the possible implications for the applicability of the Poseidon criteria [26]. They stated that FORT may reveal impaired sensitivity to FSH and should be used to guide the decision of treatment protocol, gonadotropin and stimulation doses to be used for hypo-responders.

Recently, the amount of hormone medication needed for each oocyte produced (i.e., OSI) was investigated in a retrospective cohort study that included more than 1200 women undergoing IVF with FSH/hMG stimulation [27]. Consistent with previous results in younger women with excellent pregnancy potential [8], OSI was also found to be predictive of pregnancy and live birth in older women with a more unfavorable prognosis [27]. The authors concluded that OSI could be employed in counseling women of advanced age about their reproductive potential, bridging the gap between the purely quantitative aspect of ovarian reserve and the more qualitative approach of ovarian competence.

So far, no studies have been published on whether these surrogates offer significantly higher performance than their constituent parameters, particularly in women of advanced age. To be specific: are FORT and FOI more predictive of baseline AFC, the number of pre-ovulatory follicles and the number of oocytes retrieved? As for OSI, is it more informative than the number of oocytes and the total dose of FSH used?

Despite the limitations of a retrospective study, our results seem to scale back the capabilities of FORT, FOI and OSI to answer the above questions. The combination of multiple indicators of ovarian reserve and ovarian response to COS (FOI and FORT) or in terms of FSH administered (OSI) does not seem to be more advantageous than the traditional predictors of IVF outcomes. In addition, most of these predictive factors only become available after at least one IVF cycle is conducted, limiting their usage in clinical decision-making about starting IVF. Thus, there is still a knowledge gap regarding the possibility to predict oocyte quality, especially using only information that is available before starting COS. 

In a cohort of couples with unexplained infertility and advanced female age, only age was found to be a clear predictor of live birth after IVF, further confirming the intrinsic awareness of every IVF expert: the age of the egg is what really matters. This conclusion is in agreement with what has already been reported in previous studies [23,28,29].

Indeed, numerous factors are implicated in the final outcome of an IVF treatment: oocyte and embryo quality, endometrial receptivity, women’s general health conditions, etc. It is likely that only female age is able to coherently capture all these factors. The impact of age per se seems truly relevant, considering that its correlation with IVF outcome is stronger than that of all markers considered or even combined, in spite of a rather narrow distribution of ages.

Note that FORT, FOI and OSI can be considered markers that indicate different aspects of the response to COS; we merely showed that their association with live birth after IVF is negligible when considering female age and their constituent parameters. 

This result could have more than one explanation. First of all, one should consider that each of the parameters used for the calculation of the three metrics are operator dependent. The follicle count, which registered a reduction in inter-individual variability with the introduction of three-dimensional (3D) technology [30], is still widely evaluated in 2D in most centers, as in the present study. Such a variable is inevitably affected by the skill and the accuracy of the operator. Similarly, there is no unanimous agreement on what should be the starting dose of gonadotropins in an IVF treatment [31,32]. Although several algorithms have been developed over the years [33,34], the starting dose is still widely established on the basis of the operator’s clinical sensitivity with respect to multiple parameters (AMH, AFC, age, body mass, previous COS, etc.). Obviously, the total dose of gonadotropins administered is partly affected by the initial decision. Moreover, the number of oocytes retrieved can be affected by the experience of the operator. 

Another reflection should be dedicated to the value of the ‘live birth’ as a definitive goal of an IVF treatment. This argument has been questioned from various points of view [35]. Regarding COS, several factors can interfere between OPU and a live birth. We cannot rule out that these indexes would be of different use when studied in correlation with blastulation rate, implantation rate and clinical pregnancy rate. It is likely that the correlation of these indicators with the rate of euploid blastocysts could represent a better outcome for their predictive ability. This is especially true for a sub-population of older women, where preimplantation genetic testing aneuploidy (PGT-a) appears to show the greatest benefit [36,37]. Unfortunately, our center does not perform PGT-a, and there are no previous studies [7,9,12,27] that have investigated this relationship, which is therefore worthy of further investigation in future research. 

In summary, all these parameters suffer from measurement error, offering the possibility of adding more noise, i.e., (non-)random variability. By adding more variables that are operator-dependent in a model, it is not surprising that the accuracy of these surrogates is reduced with respect to the single parameter that underlies it. This could also explain why a simple but perfect measure such as female age seems to be more informative. 

The main strength of our retrospective analysis is that it represents the first study to investigate whether a non-linear association exists between the metrics FORT, FOI and OSI and LBR in the same homogeneous population, i.e., women of advanced age with unexplained infertility, thus avoiding all the hypothetical confounding factors deriving from all the other plausible causes of infertility (male infertility, endometriosis, tubal obstruction, etc.). In addition, we carefully assessed the fit of models with the constituent parameters using contemporary statistical methods, adjusting for multiple comparisons and adjusting standard errors for couples receiving multiple cycles.

In conclusion, none of the three predictors for fertility that were proposed in the literature (FORT, FOI and OSI) were more predictive than the two ‘components’’ characteristics that were used to calculate them. This was in a population of women of advanced age (39 years or above) with unexplained infertility. Nonetheless, in light of a lack of evidence that backs up using the three metrics, we suggest that clinicians and researchers still use the components themselves (i.e., the number of oocytes collected, the number of preovulatory follicles, AFC and FSH dose) in counseling and prediction modeling.

## Figures and Tables

**Figure 1 jcm-11-00859-f001:**
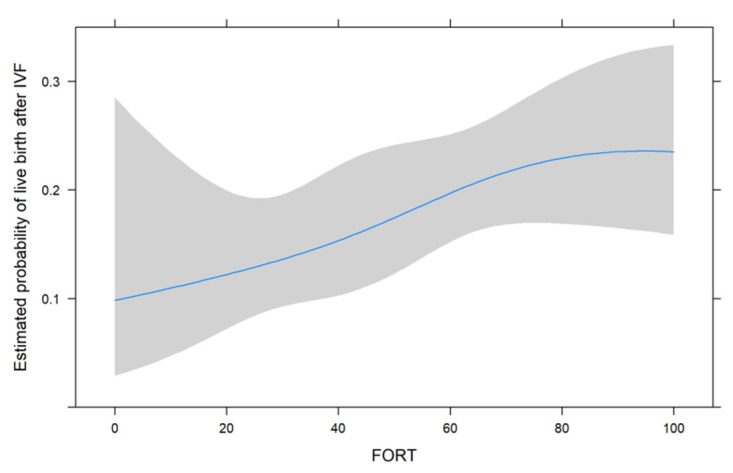
The non-linear associations between live birth and FORT.

**Figure 2 jcm-11-00859-f002:**
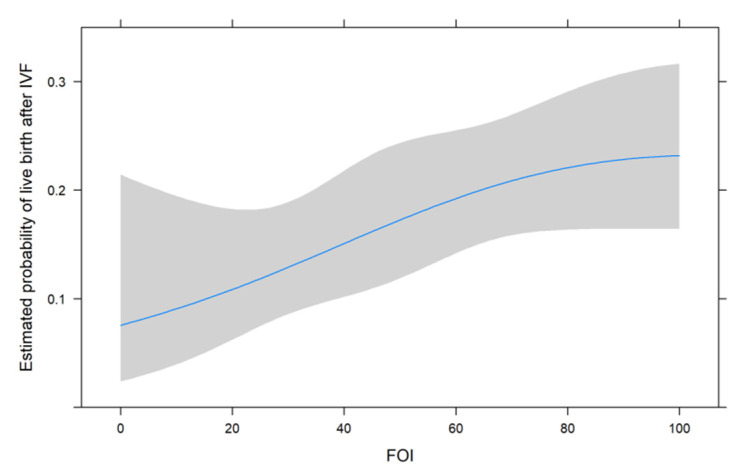
The non-linear associations between live birth and FOI.

**Figure 3 jcm-11-00859-f003:**
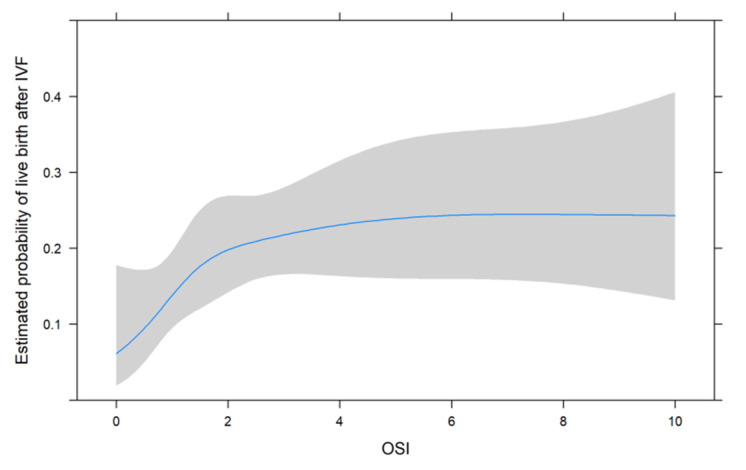
The non-linear associations between live birth and OSI.

**Table 1 jcm-11-00859-t001:** Baseline characteristics for all 590 couples.

Baseline Characteristics	
Female age at treatment (years)	40.8 (39.0−42.0)
Male age at treatment (years)	42.0 (33.0−53.0)
Duration of infertility (median, years)	2.0 (1.0−9.3)
Percentage of progressive motile sperm (median)	40 (32−48)
AMH (median)	1.1 (0.1−6.9)
FSH total dose, UI	3062 (1018−5850)
AFC (median)	10 (2−30)
OSI (median)	1.8 (0.3−11.1)
FORT (median)	50 (14−165)
FOI (median)	57 (13−175)
Number of preovulatory follicles (median)	5 (1−15)
Number of oocytes collected (median)	5 (1−18)
Infertility (primary versus secondary)	363 (62%)
Smoking (yes versus no)	69 (12%)
Poor responders (yes versus no)	326 (55%)
Previous ART	
None	238 (40.3%)
Only IUI	185 (31.4%)
IVF	167 (28.3%)

Data are presented as mean and the 2.5th–97.5th percentile, unless median is explicitly mentioned or if the characteristic is categorical or dichotomous, which is indicated by percentage signs and a reference group.

**Table 2 jcm-11-00859-t002:** Associations between patient characteristics and the odds of a live birth after IVF.

Patient Characteristics	Odds Ratio for Live Birth (95% CI)
Female age at treatment (years)	0.58 (0.48–0.72)
Duration of infertility (median, years)	0.93 (0.84–1.03)
AFC (median)	0.98 (0.94–1.03)
FSH UI, total dose	1.00 (0.98–1.02)
Number of preovulatory follicles (median)	1.06 (0.96–1.17)
Number of oocytes collected (median)	1.05 (0.94–1.17)
Infertility (primary versus secondary)	0.89 (0.58–1.37)
Poor responders (yes versus no)	0.84 (0.50–1.41)
Previous ART	
Only IUI versus none	0.74 (0.42–1.33)
IVF versus none	1.23 (0.79–1.93)
OSI (median)	0.98 (0.88–1.11)
FORT (median)	1.00 (0.99–1.01)
FOI (median)	1.00 (0.99–1.01)

Multiple logistic regression model showing associations between patient characteristics and the odds of a live birth after IVF.

## Data Availability

The datasets used and/or analyzed during the current study are available from the corresponding author on reasonable request.

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
