# Peer review of "Women in Advanced Reproductive Age: Are the Follicular Output Rate, the Follicle-Oocyte Index and the Ovarian Sensitivity Index Predictors of Live Birth in an IVF Cycle?"

_jcm, 2022, doi:10.3390/jcm11030859_

Round 1

Reviewer 1 Report

The authors performed a retrospective study evaluating the predictive value of alternative markers on live birth in women of advanced reproductive age. This paper is read with interest, the writing is clear and concise and the topic is definitely of interest.  

Author Response

We thank the reviewer for the positive comments on our manuscript.

Reviewer 2 Report

Authors presented a very interesting study on the predictability of live birth by indexes of ovarian response to controlled ovarian stimulation.

The work is well written and discussed.

However, few points deserve ameliorations.

In the introduction you mention that “polymorphic variants of the FSH-receptor” are among the factors which modulate the sensitivity of ovaries during the COS. You could take advantage from Conforti A, Vaiarelli A, Cimadomo D, Bagnulo F, Peluso S, Carbone L, Di Rella F, De Placido G, Ubaldi FM, Huhtaniemi I, Alviggi C. Pharmacogenetics of FSH Action in the Female. Front Endocrinol (Lausanne). 2019 Jun 26;10:398. doi: 10.3389/fendo.2019.00398. PMID: 31293516; PMCID: PMC6606727., to better discuss this issue too.

In my opinion a short paragraph should be dedicated in the discussion to the issue of what should be the outcome to best represent a good ovarian stimulation. Indeed, the live birth of a baby is the final and most important goal to achieve, but undoubtedly many variables may interfere with that result during the usually 9-months journey from the fertilization of the oocyte. Therefore, although I find interesting and correct both the study, its aim and its conclusion, I suggest authors to at least comment on the possibility that these indexes would be differently useful and accordingly evaluated when studied in correlation to blastulation rate, implantation rate, clinical and ongoing pregnancy rate.

Moreover, obviously the number of mature oocytes would be more informative on the chances of getting a blastocyst, or better a euploid blastocyst, and the practice of PGT-A would be of help mainly in the subpopulation of advanced maternal age for the same reasons. (La Marca A, Capuzzo M, Imbrogno MG, Donno V, Spedicato GA, Sacchi S, Minasi MG, Spinella F, Greco P, Fiorentino F, Greco E. The complex relationship between female age and embryo euploidy. Minerva Obstet Gynecol. 2021 Feb;73(1):103-110. doi: 10.23736/S2724-606X.20.04740-1. Epub 2020 Dec 11. PMID: 33306288. Shi WH, Jiang ZR, Zhou ZY, Ye MJ, Qin NX, Huang HF, Chen SC, Xu CM. Different Strategies of Preimplantation Genetic Testing for Aneuploidies in Women of Advanced Maternal Age: A Systematic Review and Meta-Analysis. J Clin Med. 2021 Aug 30;10(17):3895. doi: 10.3390/jcm10173895. PMID: 34501345; PMCID: PMC8432243.) This could be commented too, in addition to maternal age alone.

Please avoid the repetition of the terms “the two ‘components’ used to calculate them” too many times into the abstract. Try to explicit them or to change way of referring to them.

Regarding included women, in the last part of the introduction you say they are aged 38 or above, but in the initial paragraph of the methods you declare to include women aged 39 or above. Please clarify.

Any abbreviation should be spelt out at first appearance into the text (IMCS not done, methods page 3).

Tables are required to have notes below which explain how data are presented (in table 1, in the column you say “Mean or median and 2.5th−97.5th percentile” but there are also absolute number and percentage in brackets within presented results). Therefore, please change the column heading in a more generical manner and explicit below each table both how data are presented and any abbreviation used.

Author Response

We thank the reviewer for the positive comments. Thanks to his suggestions, the work has improved.

We have cited the suggested work on FSH receptor polymorphisms, which in this way can be further explored by the reader.

The live birth topic is of great interest and we fully share the reviewer's point of view. It would probably be time to think about a paradigm shift to measure the goal of a COS. Among these, probably the number of euploid blastocysts should become the real target, since the birth of a child is impacted by too many factors that an expert in reproductive medicine can only relatively impact (pre-existing uterine factors, infections in pregnancy, obstetric complications, etc); we dedicated a paragraph to this aspect in the discussion, also examining the topic relating to PGT raised by the reviewer. Certainly, this aspect will have to be addressed in future research.

The terms "the two 'components' used to calculate them" have been replaced.

We have corrected the data of "38 years", we apologize for the oversight.

All abbreviations have been mentioned.

Notes explaining the tables have been added, and the column titles corrected.